# Anthropometric Equations to Determine Maximum Height in Adults ≥ 60 Years: A Systematic Review

**DOI:** 10.3390/ijerph19095072

**Published:** 2022-04-21

**Authors:** Arnulfo Ramos-Jiménez, Rosa P. Hernández-Torres, Isaac A. Chávez-Guevara, José A. Alvarez-Sanchez, Marco A. García-Villalvazo, Miguel Murguía-Romero

**Affiliations:** 1Departamento de Ciencias de la Salud, Instituto de Ciencias Biomédicas, Universidad Autónoma de Ciudad Juárez, Ciudad Juárez 32310, Mexico; isaac.chavez@uacj.mx; 2Facultad de Ciencias de la Actividad Física, Universidad Autónoma de Chihuahua, Ciudad Juárez 32310, Mexico; rhernant@uach.mx; 3Facultad de Medicina, Universidad Autónoma del Estado de México, Toluca 50000, Mexico; josearturoalvarezsanchez@gmail.com; 4Licenciatura en Medicina, Instituto Universitario de Ciencias Médicas y Humanisticas de Tepic Nayarit, Tepic 63190, Mexico; mgarcia1901113@inumedh.edu.mx; 5Instituto de Biología, Universidad Nacional Autónoma de México, Mexico City 04510, Mexico; miguel.murguia@ib.unam.mx

**Keywords:** regression models, geriatric, bone length, genetic, epigenetic

## Abstract

Although it is common to measure bone lengths for study, methodological errors in data measurement and processing often invalidate their clinical and scientific usefulness. This manuscript reviews the validity of several published equations used to determine the maximum height in older adults, since height is an anthropometric parameter widely employed in health sciences. A systematic review of original articles published in the English, Spanish, or Portuguese languages was performed in PubMed, ScienceDirect, EBSCO, Springer Link, and two institutional publisher integrators (UACJ and CONRICYT). The search terms were included in the metasearch engines in a combined way and text form using the Boolean connectors AND and OR {(Determination OR Estimation OR Equation) AND Height AND (Elderly OR “Older adults”)}. Eleven manuscripts were selected from 1935 records identified through database searching after applying the following criteria: (1) original articles that designed and validated equations for the determination of height by anthropometric methods in adults 60 years of age and older and (2) manuscripts that presented robust evidence of validation of the proposed regression models. The validity of the reported linear regression models was assessed throughout a manuscript review process called multi-objective optimization that considered the collection of the models, the prediction errors, and the adjustment values (i.e., R^2^, standard error of estimation, and pure error). A total of 64 equations were designed and validated in 45,449 participants (57.1% women) from four continents: America (85.3%, with 46 equations), Asia (8.1%, with 10), Europe (4.6%, with 7), and Africa (2.0%, with 1); the Hispanic American ethnic group was the most numerous in participants and equations (69.0%, with 28). Due to various omissions and methodological errors, this study did not find any valid and reliable equations to assess the maximum height in older adults by anthropometric methods. It is proposed to adjust allometric mathematical models that can be interpreted in the light of ontogenetic processes.

## 1. Introduction

Along with age and body weight, height (measurement of a person from the feet to the vertex of the head) is an essential anthropometric parameter to consider in most epidemiological and public health studies. Based on these three parameters, pharmacological, nutritional, and physical training treatments are adjusted and the nutritional status and the evolutionary processes of growth and development in people and populations are determined [1,2]. Height decreases rapidly after 50 years at an annual rate of between 0.08% and 0.10% for males, and 0.12% and 0.14% for females [3], accentuating drastically after the age of 70 [4]. Maximum height is hereditary but is affected by lifestyle, environmental conditions, malnutrition, sedentary lifestyle, and social inequities [5,6,7]. On the other hand, the decrease in height is mainly associated with the changes that occur during aging (reduction of the plantar arch, the increased curvature of the spine, and the flattening of the intervertebral discs), postural habits, injuries, and diseases that affect the joints and muscle-skeletal system [3]. It has been indicated that the maximum height reached in adults helps to explain the nutritional status and diseases that occurred in the first years of life and warns of possible chronic diseases in old age [8]. In addition, a close relationship has been found between inherited short stature and the presence of genes related to atherosclerosis [9], which may mean that within a population, people with short stature have adverse lipid profiles compared to their taller counterparts. Height is also included in obesity indices (BMI) and cardiometabolic risk profiles (waist/height ratio), together predicting the occurrence and severity of the diseases [10,11].

In short, maximum height in adulthood is an evolutionary, anthropological, political, cultural, social, economic, and public health parameter that predicts chronic diseases and helps to interpret them. However, due to technical impossibility, diseases, musculoskeletal injuries, postural deviations, and aging, it is not always possible to obtain [12], making it a variable of little use in the clinic. On the other hand, when it is calculated by other means, such as the knee height (KH), tibia length, or arm span, among others, the problem increases due to the lack of a standardized method with errors and a minimal effect on its prediction [12].

To date, we have various equations (eqs) to calculate the maximum height in older adults from accessible anthropometric variables; however, many of them lack validity and reliability or are compromised to different degrees by methodological errors and inappropriate statistical treatments of the data [13,14,15]. For example, Lima et al. [16], in a sample of 168 participants over 60 years of age, compared the adequacy of 23 published equations to calculate the maximum height in the Brazilian population. Some of these equations had prediction errors between 9 cm and 12 cm. Due to the abovementioned problems, only twelve were adjusted adequately and useful as prediction models in men (R^2^ and ICC ≥ 0.7) but not in women. On the other hand, a strange and reduced systematic review [17] mentioned having analyzed 20 manuscripts concerning the association of height with various anthropometric variables, including arm span, KH, sitting height, forearm length, demi span, bi-axillary length, humeral length, hand length, thigh length, foot length, weight, standing height, half arm span, and leg length. They concluded that arm span was the most reliable anthropometric length to determine height; however, the authors did not present the respective studies and analyses or evidence to support their conclusions.

As we observed, and to the best of our knowledge, we did not find a systematic review where the validity and reliability of the regression models of the published equations were analyzed. Nor has the precision reached by methods for predicting maximum height using anthropometric methods been analyzed. Therefore, this work has two purposes: First, to make a comprehensive systematic review of manuscripts where equations were designed to predict the maximum height by anthropometric methods in adults ≥ 60 years, as well as to analyze the validity and reliability of these equations through compliance with various methodological criteria for data acquisition and analysis, a method called multi-objective optimization [18,19]. Secondly, to analyze the level of precision with which said proposed equations estimate height.

## 2. Materials and Methods

### 2.1. Information Sources

For the study design, search, analysis, and selection of the manuscripts, a team of six researchers was formed who worked independently throughout the process, except in the conclusions where they worked collegially, following the PRISMA 2020 method [20]. The selected keywords were determination, estimation, equation, height, elderly, and “older adults”. These keywords were included in the metasearch engines in a combined way and text form, using the Boolean connectors AND and OR {(Determination OR Estimation OR Equation) AND Height AND (Elderly OR “Older adults”)}. Manuscripts were searched in PubMed, ScienceDirect, EBSCO, Springer Link, and two institutional publisher integrators (UACJ and CONRICYT). The search for the manuscripts was concluded in December 2021.

### 2.2. Eligibility Criteria

The selection criteria for the manuscripts were:Publication date: without date—2021,Original articles that designed and validated equations for the determination of maximum height by anthropometric methods,Age of participants ≥ 60 years,Manuscripts where the methodology and results can be interpreted in the English, Portuguese, or Spanish languages,Manuscripts with presented evidence of validation of the proposed regression models.

### 2.3. Selection Process

In the first search, 1935 manuscripts were extracted by title from the indicated databases and were incorporated into the Zotero bibliographic reference manager. The manuscripts were ordered by title, and 383 repeated articles were eliminated, leaving 1552. Subsequently, according to the inclusion and exclusion criteria, 966 manuscripts were selected by title (586 removed). The abstracts of the selected 966 manuscripts were analyzed, eliminating 850 because they were not relevant to the manuscript’s purpose, leaving 116 of them for full-text analysis. The cites of these 116 manuscripts were also reviewed by title and abstract, including one more located in the citations. In the end, 106 of these 117 manuscripts were excluded because they were not relevant, leaving 11 of them that met the inclusion and validation criteria (Table 1 and Figure 1).

### 2.4. Data Collection and Synthesis

The regression model, parameters, statistics, and methodology for the used validation model were extracted from each eligible manuscript. The multi-objective optimization method [18,19] was used to analyze the validity and reliability of the equations in the full-text manuscripts published; drawing up a list of published criteria to validate multiple linear regression (Table 2). Primary attention was paid to the collection of the models, the prediction errors, and the adjustment values: adjusted R^2^, R^2^, standard error of estimation (SEE), and pure error (EP), among others. The model with the highest prediction fit, the highest prediction and concordance coefficients, and the lowest prediction errors were followed if a manuscript proposed several similar estimation equations. In addition, when the difference between the regression coefficients and prediction errors was less than 1%, the model with the least number of independent variables or regressors was selected, a method known as the principle of parsimony.

The above criteria are considered according to their own data analysis, also agreeing with the specific literature [21]. However, if the parametric assumptions are not strictly adhered to, they do not affect the results enough to nullify them. Therefore, the underlying question is not whether the models are robust or whether they are valid in their real application. For a better understanding of the fit criteria see [21].

## 3. Results

### 3.1. Studies and Equations Found

Among the 116 manuscripts selected for full-text analysis, we found 18 proposed new equations for adults ≥ 60 years; 11 validated their equations and presented the validation statistics (Figure 1). The first equations we found to calculate maximum height by anthropometric methods in older adults were those by Chumlea et al. in 1985 [22] in the USA, and the most recent by Jésus et al., in Africans in 2020 (Table 3). It should be noted that the equations of Chumlea et al. from 1985 were not validated at the time, but were commonly used in various research protocols, later validated in various populations; for example, Jésus et al. [23] validates them in Africans and Lima et al. [16] in Brazilians. Table 4 summarizes a total of 64 equations, which, according to the parsimony, inclusion and exclusion criteria, estimation power, and prediction error described above, were the most straightforward and most reliable. These equations were designed and validated in 45,449 participants (57.1% women). The participants were from four continents: America (85.3%, with 46 equations), Asia (8.1%, with 10), Europe (4.6%, with 7), and Africa (2.0%, with 1) with the Hispanic American ethnic group being the most numerous in participants and equations (69.0%, with 28). Finally, Palloni and Gued [24], using the database of the Health, Well-being, and Aging survey [25] applied in Latin America and the Caribbean, present and validated 14 equations. Only the study by Jésus et al. [23], previously mentioned, does not exclude participants with postural problems.

According to the literature consulted (Table 3 and Table 4), the most recognized author was Chumlea, with 12 equations validated at three different times [22,26,27] and by different authors. Additionally, most of the data were taken from surveys, including the Massachusetts Hispanic Elders Study 1993–1997 (MAHES), the National Health Examination Survey (NHES, 1960–1970), National Health and Nutrition Examination Survey III (NHANES III, 1988–1994), and the Fashion Information and Technology Size Korea [28]. In only one manuscript, the authors present the adjusted R^2^ [29], but they do not validate their model; furthermore, not all authors present the standard estimation error (SEE) or the pure error (PE). The most involved parameter in the equations was KH in 59 equations, followed by age (38 equations). A total of 27 equations involve both; the other parameters are demi span (DS; 5 equations), arm-span (AS; 4 equations), and ulna-length (UL; 2 equations). Table 4 reports various statistical parameters, including: adjusted R^2^ (1 equation) R^2^ (30 equations), SEE, ICC (2 equations) and PE (40 equations).

**Table 3 ijerph-19-05072-t003:** Articles and studies where the analyzed anthropometric equations (eqs) are reported.

Id Article	Article	Study
1 *	Bermúdez et al., 1999. [30]	National Survey MAHES (Massachusetts Hispanic Elders Study 1993–1997). Random cross-validation (~50%). People with postural problems were excluded and outliers were removed.
2 *	Chumlea et al., 1985. [22]	The USA, outpatient volunteers without postural problems (people with excessive spinal curvature were excluded). Equations were widely used and validated by various authors.
3 *	Chumlea and Guo, 1992. [26]	National Health Examination Survey USA (1960–1970). Cross and secular validation for 30 years. Non-institutionalized people.
4 *	Chumlea et al., 1998. [27]	Third National Health and Nutrition Examination Survey (NHANES III 1988–1994). Cross and secular validation.
5 *	Hwang et al., 2009. [31]	National survey on people without bone or joint problems. Cross-validation 80–20% and external. Extreme data were excluded.
6 *	Jésus et al., 2020. [23]	EPIDEMCA (Epidemiology of Dementia in Central Africa). People with joint and postural problems were included. Cross and convergent validation vs. Chumlea 1992.
7 *	Jiménez-Fontana and Chaves-Correa, 2014. [32]	CRELES national survey. Cross-validation at 50%. People with spinal deformities were excluded.
8 *	Karadag et al., 2012. [33]	Convenience study designed in adults (19–50 y) and validated in adults older than 59 y.
9 *	Lera et al., 2009. [34]	SABE survey. Cross-validation at 50% and by Lima et al., 2018 in Brazilians.
10	Malnutrition Advisory Group (MAG, 2011). [35]	British nutritional screening of adults: a multidisciplinary responsibility.
11 *	Mendoza-Núñez et al., 2002. [36]	Sample for convenience. Cross validation 50%.
12	Narančić et al., 2013. [37]	Zagreb, Croatia. Institutionalized people Survey.
13	Nguyen et al., 2021. [29]	Sample for convenience.
14 *	Palloni and Guend, 2005. [24]	SABE survey in Latin America with random sampling. 50% random cross-validation.
15	Pertiwi et al., 2018. [38]	Sample for convenience.
16	Ritz et al., 2007. [39]	Multicenter study.
17	Weinbrenner et al., 2006. [40]	Sample for convenience.
18	Zhang et al., 1998. [41]	Aleatory survey.

* Validated studies.

**Table 4 ijerph-19-05072-t004:** Regression models found in the literature to measure maximum height in adults ≥ 60 years old.

Id Article	Regression Model. Lengths (cm), Age (y)	Sample (*n*)	Country or Ethnic Group	Sex	Age (y)	Height ± SD (cm)	R^2^	SEE	PE
1 *	70.28 + 1.81 KH	128	Hispanic American	men	60–92	165.1 ± 6.2	0.72		2.8
1 *	68.68 + 1.90 KH—0.123 age	166	Hispanic American	women	60–92	152.7 ± 6.0	0.73		2.3
1 *	53.42 + 2.13 KH	81	Puerto Rican	men	60–92	164.1 ± 6.2	0.77		3.1
1 *	66.80 + 1.94 KH—0.123 age	87	Puerto Rican	women	60–92	151.8 ± 5.9	0.7		2.9
2 *	60.65 + 2.04 KH	106	Non-Hispanic white American	men	65–104	169.1 ± 6.9	0.67	3.8	
2 *	64.19 + 2.03 KH—0.04 age	130	Non-Hispanic white American	women	65–104	156.7 ± 5.6	0.65	3.5	
3 *	75.00 + 1.91 KH—0.17 age	451	White	women	60–80	156.8 ± 6.8	0.59	4.4	3.48
3 *	58.72 + 1.96 KH	60	Black	women	60–80	156.8 ± 7.1	0.70	4.06	
3 *	59.01 + 2.08 KH	438	White	men	60–80	170 ± 7.0	0.68	3.91	3.32
3 *	95.79 + 1.37 KH	50	Black	men	60–80	167.7 ± 6.2	0.51	4.18	
4 *	78.31 + 1.94 KH—0.14 age	1369	Non-Hispanic white	men	≥60	173.5 ± 6.7	0.69	3.74	3.62
4 *	79.69 + 1.85 KH—0.14 age	474	Non-Hispanic black	men	≥60	172.7 ± 6.9	0.70	3.81	3.68
4 *	82.77 + 1.83 KH—0.16 age	497	Mexican-American	men	≥60	166.9 ± 6.3	0.66	3.69	3.64
4 *	82.21 + 1.85 KH—0.21 age	1472	Non-Hispanic white	women	≥60	159 ± 6.6	0.64	3.98	3.8
4 *	89.58 + 1.61 KH—0.17 age	481	Non-Hispanic black	women	≥60	160.2 ± 6.2	0.63	3.83	3.81
4 *	84.25 + 1.82 KH—0.26 age	457	Mexican-American	women	≥60	153.2 ± 6.3	0.65	3.78	3.45
5 *	70.87 + 1.96 KH—0.14 age	596	Korean	women	20–69	152.9 ± 5.2	0.69	2.88	
5 *	74.63 + 1.95 KH—0.09 age	2020	Korean	men	20–69	169.3 ± 6.4	0.73	3.32	
6 *	72.75 + 1.86 KH—0.13 age + 3.41 sex (0: women; 1: men)	887	African	women (61.5%) and men	≥ 65	women = 152.9 ± 5.2 men = 169.2 ± 6.4	0.67	0.75	
7 *	58.28 + 2.20 KH—0.10 age	936	Costa Rican	men	≥60	163.1 ± 6.6	0.75	3.28	3.32
7 *	62.0 + 2.10 KH—0.163 age	1101	Costa Rican	women	≥60	149.1 ± 6.6	0.7	3.37	3.52
8 *	52.46 + 2.24 KH	219	Turkish	men	60–97	168.2 ± 6.1	0.78		
8 *	51.44 + 2.21 KH	219	Turkish	women	60–97	156.3± 5.3	0.88		
9 *	69.87 + 1.85 KH—0.11 age	944	Brazil	women	60–99	152.4 ± 5.2	0.58	3.58	3.8 ^ε^
9 *	67.2 + 1.96 KH—0.08 age	713	Brazil	men	60–99	165 ± 6.4	0.69	3.66	4.25 ^ε^
9 *	75.17 + 1.78 KH—0.1 age	615	Chile	women	60–99	165 ± 6.4	0.54	3.24	4.34 ^ε^
9 *	64.88 + 2.09 KH—0.1 age	389	Chile	men	60–99	164.8 ± 6.6	0.7	3.67	5.28 ^ε^
9 *	73.09 + 1.87 KH—0.19 age	607	Mexico	women	60–99	148.3 ± 6.2	0.59	4.0	4.9 ^ε^
9 *	63.88 + 1.99 KH—0.06 age	388	Mexico	men	60–99	162.5 ± 6.3	0.67	3.67	5.28 ^ε^
10	86.3 + 3.15 UL	62	White American	men	>65	169.1 ± 5.6			
10	80.4 + 3.25 UL	60	White American	women	>65	158 ± 6.9			
10	71 + 1.2 DM	67	White American	men	>55	169.1 ± 5.6			
10	67 + 1.2 DM	62	White American	women	>55	158 ± 6.9			
10	75.00 + 1.91 KH—0.17 age	229	White American	women	60–90	158 ± 6.9			
10	59.01 + 2.08 KH	229	White American	men	60–90	169.1			
11 *	52.6+ 2.17 KH	186	Mexican	men	60–97	162.9 ± 5.9	0.69	3.32	3.29
11 *	73.7+ 1.99 KH—0.23 age	550	Mexican	women	60–97	149.3 ± 5.9	0.74	2.99	2.98
12	98.50 + 1.755 KH—0.350 age	234	Croatian	women	85–101	152.7 ± 6.0	0.52	4.4	
12	56.72 + 2.091 KH	80	Croatian	men	85–101	167.8 ± 7.0	0.6	4.5	
13	59.06 + 2.12 KH	269	Vietnamese	men	18–64	165.7 ± 5.4	0.67		
13	57.37 + 2.09 KH	186	Vietnamese	women	18–64	155.1 ± 5.6	0.64		
14 *	94.1 + 1.21 KH	4898	Hispanic	women	≥60	153.3 ± 7.8			7.08
14 *	98.2 + 1.29 KH	3139	Hispanic	men	≥60	166.4 ± 7.8			6.93
14 *	101.8 + 1.06 KH	4269	Hispanic black	women	≥60	154 ± 7.7			6.87
14 *	105.6 + 1.16 KH	2725	Hispanic black	men	≥60	167.1 ± 7.9			7.12
14 *	88.5 + 1.32 KH	319	Hispanic mestizo	women	≥60	151 ± 6.7			5.32
14 *	67.2 + 1.88 KH	170	Hispanic mestizo	men	≥60	164.3 ± 7.5			4.36
14 *	62.6 + 1.81 KH	629	Hispanic Mexican	women	≥60	148.5 ± 6.7			5.29
14 *	59.6 + 1.99 KH	414	Hispanic Mexican	men	≥60	162.3 ± 6.7			5.75
14 *	109.0 + 0.91 KH	511	Hispanic mulatto	women	≥60	154.4 ± 7.6			7.49
14 *	108.9 + 1.08 KH	271	Hispanic mulatto	men	≥60	166.3 ± 7.5			6.37
14 *	82.9 + 1.43 KH	2583	Hispanic non-white	women	≥60	153.9 ± 8.4			7.82
14 *	87.5 + 1.48 KH	1623	Hispanic non-white	men	≥60	166.2 ± 8.2			7.28
14 *	110.8 + 0.87 KH	2114	Hispanic white	women	≥60	152.6 ± 7.1			6.63
14 *	112.8 + 1.03 KH	1515	Hispanic white	men	≥60	166.7 ± 7.4			7.03
15	40.915 + 0.457 AS + 0.818 KH	71	Indonesian	women	60–69	157.0 ± 6.92	0.98 ^ε^		
15	34.426 + 0.513 AS + 0.813 KH	65	Indonesian	men	60–69	145.4 ± 5.78	0.99 ^ε^		
16	90.20 + 1.538 KH + 5.96 sex (0: women; 1: men)—0.094 age	752 (50.4% women)	France non-Hispanic Caucasian	women and men	≥54	men: 170.6 ± 6.8. women: 157.7 ± 5.9	0.77	4.4	
17	77.821—0.215 age + 1.132 DM	271	Spain	men	≥65	163.1 ± 6.4			
17	88.854—0.692 age + 0.899 DM	321	Spain	women	≥65	150.0 ± 5.2			
18	67.78 + 2.01 KH	130	Chinese	men	30–90	163.2 ± 5.5	0.59	4.07	
18	39.56 + 0.75 AS	130	Chinese	men	30–90	163.2 ± 5.5	0.69	3.55	
18	78.46 + 1.79 KH—0.066 age	117	Chinese	women	30–90	151.5 ± 5.2	0.56	4.01	
18	38.21 + 0.76 AS	117	Chinese	women	30–90	151.5 ± 5.2	0.71	3.03	

id Article as in Table 3. AS = arm span, DS = demi span, KH = knee height, UL = ulna length.* = Studies with validated equations. ^ε^ = Most likely wrong values. PE = pure error, SEE = standard error of estimation, R^2^ = predictive power.

### 3.2. Accuracy of Reported Equations

To analyze the precision of the equations found in greater depth, we limited ourselves to studying the most frequent ones with the same structure or form:

The equations of the form:$$\mathrm{height}=\beta o+\beta_{1}\;\cdot \;\mathrm{KH}+\varepsilon$$
are the most frequently reported in the articles selected in this analysis, with a total of 28 equations (10 for women and 18 for men, Table 5) whose R^2^ values are on average 0.67 (women) and 0.68 (men). This form of eq means that the person’s maximum height can be calculated by adding to a constant, a ratio of KH. (Table 5).

When graphing this family of equations, it can be seen that they are distinguished from each other by slight variations in the slope and the ordinate to the origin (Figure 2) and that they mostly intersect in the values that are expected to be the most frequent combinations of height vs. KH in older adult populations. It can be seen that the most significant number of intersections is in the approximate range of 40 to 50 for women and 45 to 55 for men (blue ellipses in Figure 2). Thus, it can be suspected that the variations in these families of equations are due to bias in the participant samples and variations of a universal allometric relationship.

When the various versions are analyzed together in this way, it is found that:
-The average of R^2^ is 0.67 for eqs of women and 0.68 for eq of men (Table 5).-When plotting the slopes against the y-intercepts, a straight line with R^2^ of 0.99 is formed, both for eqs for women as well as men (Figure 3).-In eqs with more than 1500 participants, the intercept is closer to unity (symbols in red in Figure 3), and they are never greater than 1.5. A total of 90% of the equations with slopes greater than 1.5 were derived from less than 500 participants’ samples.

Gender (female/male) is an essential variable since the allometric relationship between height and KH is more similar between people of the same gender and a different ethnic group than between people of different gender and the same ethnic group.

The mean coefficients of variation (SD divided by the mean) for height are 4.8% and 4.2% for women and men, respectively (Table 5). That is, there is little variability in the participants’ maximum height in the studies from which the equations were derived. This data is crucial because it may serve as a parameter for comparison with the samples of participants from other studies.

## 4. Discussion


What do the results mean?


One of the findings of this study is the frequent relationship of KH in the equations to estimate height, where at the same time, we find a linear and allometric relationship between these two variables. In this sense, we consider whether the relationship between these two measures will be genuinely linear. Alternatively, perhaps the bias in the study objectives (estimation of stature, and not the explanation of the relationships between the dimensions of the parts of the human body) ignores the “power law” proposed in the zoological context a century ago [42].


2.Highlighting its clinical importance.


The importance of establishing mathematical models for estimating maximum height based on body parts is for two contexts, clinical and public health. From the clinical point of view, having precise models will allow us to appreciate deviations from biological and mathematical normality due to genetic and epigenetic effects, thereby evaluating the patient’s health status more precisely. From the point of view of public health, the models that estimate this type of anthropometric relationship will make it possible to evaluate, through comparisons between different populations of the same ethnic group, the consequences of undergoing different lifestyles, including the type of diet and/or physical activity, among others.


3.In what way and why are the results similar or different between the different authors?


The main points of agreement between the various studies analyzed are (a) that the KH parameter is the one that presents the highest correlation between the various body measurements published; (b) that to obtain maximum correlation values, the analysis should be done separately for women and men; and (c) without proposing the allometric context, the proposed equations seem to be first-order linear, except for a few cases
4.What does this study add to science?

This study provides a solid example of how anthropometric studies can argue their validity by support with the appropriate statistical methods. In this analysis, an orderly and detailed synthesis of the equations reported in the literature derived from analyses that meet statistical validity criteria is made. For the case of the linear equation between height and KH, values of the parameters βo and β1 are proposed, which may be refuted, adjusted, or rejected in future studies, but the most important thing is that they can be used as points of comparison.


5.What are the strengths and weaknesses of this study?


One of the strengths of this study is the systematic form and the explicit criteria for the selection of manuscripts, thus excluding from the analysis equations derived from studies with methodological errors. One of the weaknesses is that the analysis could not be done specifically for the various ethnic groups, and is only differentiating between women and men, due to the lack of validated studies. Another weakness is that we only found first-order linear equations, with KH and age as the most determining variables, lacking an equation that integrates both the genetic and epigenetic relationships of the various body segments. The last one is a better idea of a human development study from the anthropometric point of view.

Within physical anthropology, lengths, widths, circumferences, depressions, edges, and other physical bone forms, are parameters widely studied since ancient times (paleoanthropology). They provide us information about gender, growth, development of the different organs and systems, and evolution of the human species in social, cultural, nutritional, migration, ethnic, and current genetic aspects [43]. Additionally, within biomechanics and bioengineering, bone lengths such as standing height, sitting height, arm, and leg length, along with skeletal joints and muscles, are addressed for the study of movement, ergonomics, and mechanical efficiency. All the body segments present a relative geometric proportion in an individual [44]. This critical aspect allows us to validate the studies considering ethnic, gender, and nutritional factors. In this sense, the importance of this work and that of standardizing anthropometric measurements is to reduce systematic and measurement errors, thereby making it possible to compare the results between different studies and populations.

### 4.1. Equations Reported in the Literature

The most straightforward and most practical equations are examined in this manuscript, with the lowest prediction errors (SEE), the highest regression coefficients (R^2^), and the best fits according to the regression model evaluation criteria described in Table 2. However, considering the other manuscripts chosen for a full-text reading, we observe that, in general, not all the manuscripts where equations are designed had the purpose of testing their validity. For example, an eight-year longitudinal study [41] elaborated eight equations in 247 Chinese (30–90 y, 47% women). They found that, compared to height and arm span (R = −0.21 vs. −0.19 and R = −0.27 vs. −0.26; for men and women, respectively), KH does not change substantially with age (R = −0.06 vs. −0.15 for men and women, respectively), making KH a reliable parameter to determine the maximum height in older adults. Table 3 and Table 4 present the different studies where equations are used to calculate the maximum height in older adults. In thirty-two equations from nine studies, the authors do not report compliance with the minimum assumptions to run a linear regression model (Table 4) as a linear relationship between the dependent and independent variables, normality, homoscedasticity, independence of errors, and non-multicollinearity, among others mentioned in Table 2. Strictly speaking, we consider that, as in all types of research, when researchers think of using regression models, they must previously establish the error levels (e.g., mean error and its confidence interval) and adjustment accepted in their models before carrying out any measurement. However, we observe that this is not done with the various parameters, and they support their claims with the most common parameters or ones that best justify their hypotheses.

It is important to note that the reader will encounter considerable variance in the regression equations’ pure error (PE) and standard error of estimate (SEE). In a good regression model, PE and SEE are very similar, with PE being slightly smaller. Both represent the determination error when applying the regression equation to the sample. The lower the PE and the lower the SEE, the higher the validity and reliability of the equation. It is proposed that models with a PE greater than ± 2.5% determination have a considerable negative effect when predicting the response. However, readers may have a different opinion depending on the type of study, purpose, and study population. So, here we present the PE of the equations where the authors included it and the mean of the measured height so that the reader calculates the percentage, they consider acceptable in the chosen equation. Additionally, in the validation processes, besides providing the prediction and adjustment values of the model, the authors must present at least the R, paired T-test, ICC, and especially Lin’s CCC values. The last one is greater than 0.95; however, all those mentioned above are fulfilled in very few cases, especially in small samples.

Another problem with the manuscripts reviewed in this study is the misunderstanding and application of statistical methods. For example, Fogal et al. [45] and Cape et al. [46], contrary to what they conclude, find the equations of Chumlea [20], using the height of the knee, and those of the Malnutrition Advisory Group (MAG) [35], using the length of the ulna, are very suitable for estimating the maximum height in the Brazilian population over 60 years old. The Chumlea equations overestimate maximum height by 1.2 cm, that is, 0.8% (158.9 ± 9.1 cm vs. 160.1 ± 7.6 cm, for the measured and the predicted, respectively); in the same sense, the equations of the MAG [35,47] underestimate it by 0.033 cm, that is, 0.2% (161.07 ± 8.77 cm vs. 160.74 ± 7.48 cm for the measurement and the predicted one, respectively). To support their conclusions of invalid equations, Fogal et al. [45] and Cape et al. [46] erroneously take the statistic *p*-value of the Student’s *t* instead of the effect size or adjustment indices, such as Cohen’s d, adjusted R^2^, ICC, and CCC, among others. Ideally, when regression models are presented, the estimators should be accompanied by the errors and confidence intervals, in both the general model and each one of the estimators of the equation: Y = Bo +B_1_X_1_ + B_2_X_2_ + … + BnXn + ε.

In summary, the most common problems found in the 116 manuscripts consulted are: the authors do not present the entire methodology used for the studies presented, the equations do not validate them, equations and descriptive statistics are presented without mentioning or presenting the internal and external validation statistics of the regression models, the authors do not show the prediction errors or check the assumptions for the regression analyses, and they are not representative enough to support their hypotheses. Therefore, people who wish to use predictive equations must first ensure that these equations have been validated in a population such as the one studied and present clinically useful SEE. Second, these equations must be revalidated and applied again to a sample of the population to be studied, and the adjustment level of the estimators and pure errors calculated.

### 4.2. Estimation of the Parameters of Linear Equations

The high correlation (R^2^ = 0.99) between the slope and the intercept of the linear models (maximum height = βo + β_1_ · KH + ε) suggests that the βo and β_1_ variations are largely due to biases in the samples. As the sample size increases, it is expected that the parameters βo and β_1_ of the linear relationship will tend to have more similar values between different populations, even in different ethnic groups. In addition, the graph in Figure 3 suggests that the value of β_1_ will be less than 1.5, and the ordinate value to the origin, βo, will be greater than 80 cm. In this way, if both parameters are averaged for the equations with samples greater than 1500 participants, the resulting equations are:
-women: maximum *height* = 97.40 + 1.14 · KH + ε-men: maximum *height* = 101.03 + 1.24 · KH + ε

### 4.3. Intrinsic Allometric Variability of the Human Being

The low R^2^ values of the above-mentioned equations support the hypothesis that human beings have an allometric intrinsic, intraspecific variability between height and KH. Maximum height has an intrinsic variability since, in the data set of the studies included in this analysis, it is estimated that maximum height has a variability between 4–5% (measured by the coefficient of variation, Table 5). Thus, the allometric relationship of height vs. KH has variability throughout the population, represented by the low R^2^ coefficients (on average 67% in women and 68% in men). In other words, the prediction of the maximum height based solely on the KH parameter has limits imposed by the variability of the human being, even within the same ethnic group. Thus, the search for equations that estimate maximum height based on different body measurements should consider more than one parameter. However, the results of this analysis also support that KH is a helpful measure in equations to estimate height, but in combination with other lengths.

### 4.4. Towards the Search for More Precise Allometric Relationships

Allometry refers to changes in the relative dimensions of body parts that correlate with changes in overall size [48]. The study of allometry dates back more than a hundred years to zoologists and mathematicians, but it was in the 1930s that important concepts were consolidated that would later support the study of the evolution of species. One of the critical contributions of the time was what is known as the “power law” [42], a mathematical model that is expressed through the formula:y = b · x^α^(1)
where y is the size of one of the organism’s parts, x is the total size of the organism, and b and α are constants. The constant b is a scaling factor that expresses the differences in size between comparable organisms (of the same shape) with the same α. The constant α is the ratio of specific growth rates of y/x. It is important to note that this model tries to explain the ontogenetic process of growth and not precisely estimate the size of the organism’s entire body based on the size of any of its parts. Another important aspect is the dynamic conception of allometric relationships, where the constant α in the equation refers to the concept of “constant differential growth ratio,” and not to the relationship between a part of the body and its total size, but to a ratio between two or more variables to determine growth-rates (y = βo + β_1_x_1_^α1^ + β_2_x_2_^α2^ + … + β_n_x_n_^αn^ + ε, where for lineal regression ^α^ = 1). This model emphasizes understanding species’ evolution, supporting the hypothesis that proportionate “dwarf” and “giant” species often owe their status to changes in a single gene that controls simple hormonal mechanisms. This model has also made it possible to establish the concept of allometric rearrangements (different b and same α) and to propose that these can result from simple mutations [49]. Thus, the models presented in Figure 2 and Figure 3 can be interpreted under the inspiration of the light that the “power law” can illuminate. For example, return to the concept of allometric transposition. Sexual dimorphism in humans has been the subject of various studies; for example, dimorphism in pelvic height-width ratio and iliac blade orientation was found to be mediated by a growth hormone (GH) secreted at puberty by females [50]. In this way, as already mentioned, sexual dysmorphism is not only in a weight-mass relationship, but there are also allometric differences between other body parts which makes it necessary to analyze women and men separately in the construction of the anthropometrics equations to predict maximum height.

Although the equations analyzed in this study aimed to estimate height, they are assigned a predictive value. In this sense, it is essential to consider them as understanding ontogeny models of the *Homo sapiens* anthropometric variability. If ever more accurate predictions are desired, understanding ontogenetic processes will help with that task. For this reason, in the next section, we discuss the role of genetics in bone growth.

### 4.5. Bone Growth and Genetics

Bone growth is highly regulated by gene expression and is a variable that has not been included in studies such as the ones reviewed here, increasing the prediction error. In this sense, only the mutation of the single nucleotide polymorphism (SNP) of one of the membrane proteins belonging to the superfamily of G protein-coupled receptors (GPCRs), (Figure 4. https://www.uniprot.org/uniprot/Q86SQ4; accessed on 5 March 2022) Adgrg6, also called GPR126 [51], modifies bone length through the formation of the second messenger adenosine 3′,5′-monophosphate (cAMP). The affectation in the peptide sequence called the Stachel sequence in the amino terminal [52], within the ectodomain of GPR126 that functions as an agonist anchored to activate cAMP signaling, decreases osteoblast proliferation [48]. The activation and inhibition of these GPCRs are regulated by administering commonly used drugs, for example products for reducing body weight such as Forskolin, and steroidal anti-inflammatory drugs such as glucocorticoids [53]. These drugs downregulate osteoblast differentiation and increase osteoclast differentiation [43]. Another importance of these membrane GPCRs is that 3% to 5% of human genes encode them and between 20% and 30% of drugs for clinical use and drugs for legal or illegal use, such as opiates, cannabinoids, and serotonergic act on them [51]. However, the effect size of the relationship between changes in the GPR126 polymorphism with maximum height has not been established.

## 5. Conclusions

Strictly speaking, the manuscripts reviewed here lack the appropriate methodology to be published, establishing the need for the authors to carry out complete methodological processes to validate their equations and determine with sufficient certainty the maximum height by easy and accessible anthropometric lengths, especially in adults aged 60 years and over. In other words, we generally observed abundant errors in the measurement and statistical processing of the data. Therefore, we did not find any equation that meets the necessary validity and reliability requirements to determine the maximum height by different anthropometric methods in adults aged 60 years and over. The small size of several samples of participants in which the anthropometric measurements were made produces significant biases in the parameters of the derived equations. We consider that the ontogenetic processes that originate the human allometric variations should be included in the models to estimate height.

## Figures and Tables

**Figure 1 ijerph-19-05072-f001:**
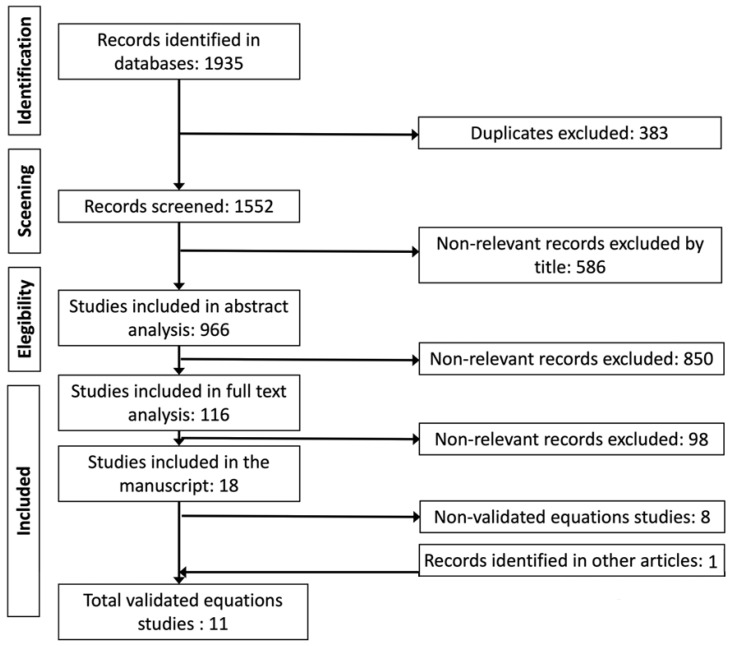
PRISMA 2020 method for the manuscript selection (Page et al., 2021 [20]).

**Figure 2 ijerph-19-05072-f002:**
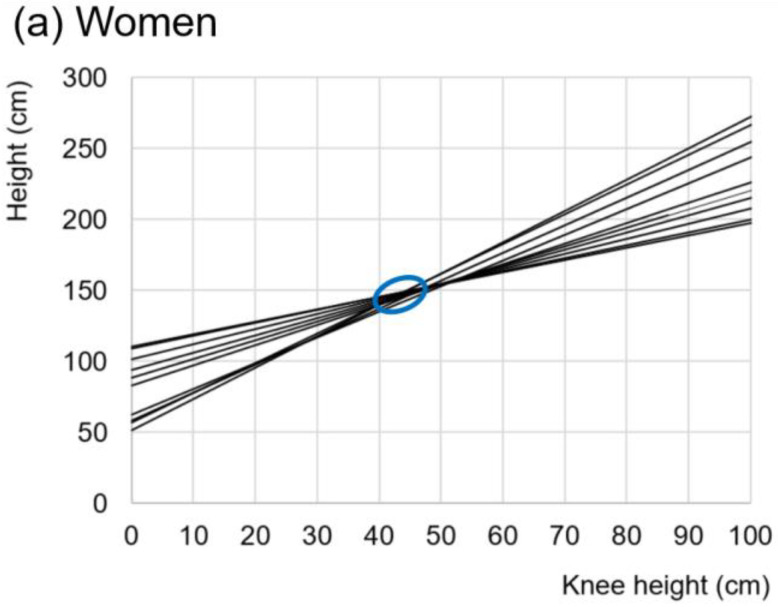
Family of straight lines of eqs of the form height = βo + β_1_ · KH + ε. The blue ellipses indicate the area of intersection of the lines.

**Figure 3 ijerph-19-05072-f003:**
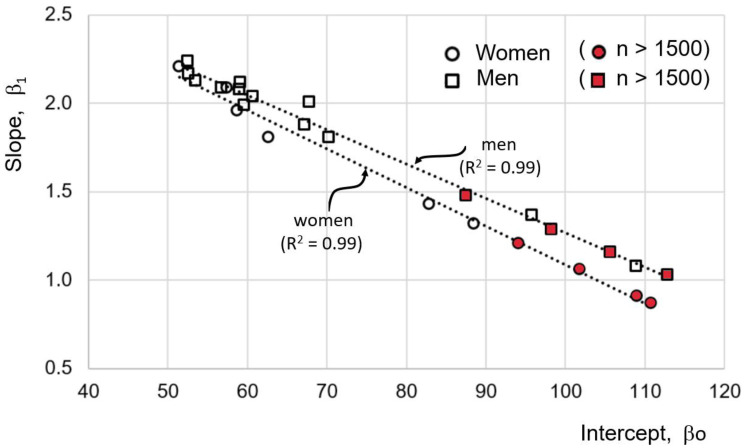
The linear relationship between the βo and β_1_ parameters of the lines in Figure 2.

**Figure 4 ijerph-19-05072-f004:**
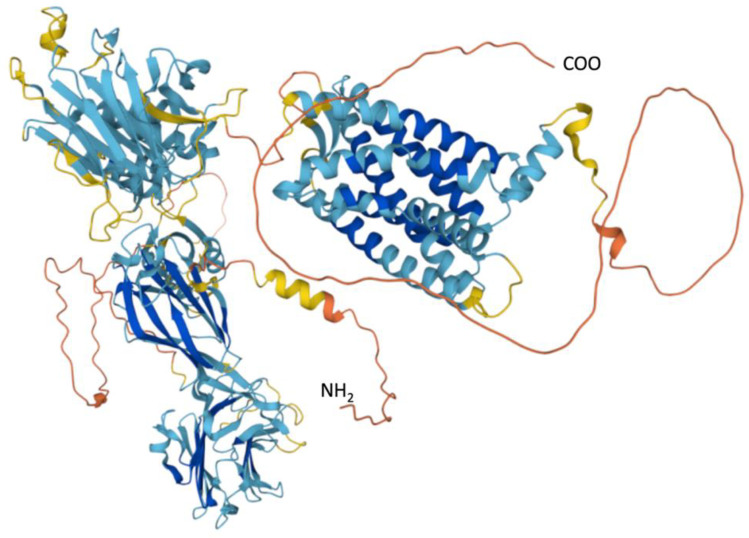
Adhesion G-protein coupled receptor G6. From https://www.uniprot.org/uniprot/Q86SQ4 (accessed on 5 March 2022; Mogha et al., 2013 [51]).

**Table 1 ijerph-19-05072-t001:** Natural history of the project and manuscript.

1. Partial search of the literature on the subject: A.R.J.
2. Encounter of a possible problem-or study opportunity: A.R.J.
3. Selection of participants: A.R.J.
4. Project design and planning: The whole team
5. Partial and independent search in the literature about the topic: The whole team.
6. Selection of the question and study hypothesis: The whole team.
7. Selection of keywords and elaboration of the syntax for the search of manuscripts in the literature: The whole team.
8. Preparation of inclusion and exclusion criteria: The whole team.
9. Exhaustive and independent search of the manuscripts in reliable metasearch engines: A.R.J., I.A.C.G., J.A.A.S., and M.G.V.
10. Creation of a database of the manuscripts found (Zotero): A.R.J.
11. Elimination of repeated articles: A.R.J.
12. Independent selection by the title of the manuscripts found and the database created in Zotero: A.R.J., I.A.C.G., J.A.A.S., and M.G.V.
13. Elimination of repeated articles: A.R.J.
14. Independent selection by the abstract reading of the selected manuscripts by title: A.R.J., I.A.C.G., J.A.A.S., and M.G.V.
15. Elimination of repeated articles: A.R.J.
16. Selection of the chosen manuscripts to complete reading of the manuscript: The whole team.
17. Analysis, elaboration of Tables, Figures, and discussion of the results: A.R.J., R.P.H.T., and M.M.R.
18. Preparation of the final manuscript: A.R.J, R.P.H.T, and M.M.R.

Initials = participating researcher.

**Table 2 ijerph-19-05072-t002:** Criteria to evaluate the validity and reliability of the regression models.

**Validity Criteria (Accuracy)**
1. Provide a clear and complete description of the methods and procedures.
2. Use of valid and reliable instruments for data collection. If necessary, mention the calibration processes of the instruments.
3. Use of standardized measurement procedures.
4. Technical training in anthropometrics.
5. Randomization and sample size: In this work, we consider an *n* ≥ 100 and 10 more for each independent variable added to the model; the previous is to favor the central limit theorem or normal distribution of the data.
6. Report of measurement errors: Technical measurement error (TEM). Intraclass correlation coefficient (ICC).
7. Internal validation analysis or cross-validation (generally 50–50% or 80–20% in small populations) and external validation of the model or independent validation (≥50).
**Reliability Criteria (Precision)**
1. Use of normal distribution of the data for each variable in the model.
2. Elimination or correction of outliers and/or transformation of the data. Interquartile range: ±2.2 times the interquartile difference = Q1 and Q3 ± 2.2 (Q3-Q1). Z-score ≤ 2.5 SD. Cook’s distance < 1. Mahalanobis distance values less than 0.001.
3. Make data transformation in case of outliers cannot be removed or corrected. The data transformation commonly homogenizes the database and makes its estimates more robust; e.g., logarithm, root, power, or exponents transformations normalize the data, remove outliers, and randomize the residuals.
4. Linearity between the dependent and independent variables. Plot the raw data between them and observe their kinetics; if necessary, make transformations.
5. Homoscedasticity or constant variance of the residuals.
6. Theoretical coherence of the associations: expected signs and relevant variables present in the model.
7. Independence of errors or residuals.
8. Normal distribution of errors or residuals.
9. Non-multicollinearity.
10. Determination coefficients: R^2^ and adjusted R^2^, plus their confidence intervals. The last two, especially if they are two or more independent variables.
11. Hypothesis test for the general model and the independent variables: generally, *p* < 0.05.
12. Model goodness-of-fit criteria. Standard Error of Estimation (SEE) or Square Root of the Mean Square Error RMSE=∑ [(Yi−Y^i)2](n−p−1)Pure error=∑ [(Yi−Y^i)2]n Where *n* is the number of participants and *p* is the number of variables in the model.
13. Degree of agreement or concordance between the measured value and that estimated by the model: Pearson’s correlation ≥ 0.8 Paired T or Wilcoxon. Intraclass Correlation Coefficient ICC ≥ 0.7, Lin’s Concordance Correlation Coefficient (CCC ≥ 0.95), Graph of measured vs. predicted values. Coefficients: R^2^ = R^2^ predicted, Beta close to 1 and constant or y-intercept close to 0. Make a Bland-Altman plot of the differences between measured and predicted vs. its average: random distribution with mean 0 and constant variance.
14. Have in mind the principle of parsimony, simplicity, and economy.
15. Carry out the inclusion of confounding factors in the models.

**Table 5 ijerph-19-05072-t005:** Totals and averages of parameters of eqs of the family maximum height = βo + β_1_ · KH + ε.

Sex	Women	Men
No. Equations	10	18
Total participants	15,937	11,394
Mean R^2^	0.67	0.68
Mean CV (Height)	4.8%	4.2%

## Data Availability

Not applicable.

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
