# Peer review of "Anthropometric Equations to Determine Maximum Height in Adults ≥ 60 Years: A Systematic Review"

_ijerph, 2022, doi:10.3390/ijerph19095072_

Round 1
Reviewer 1 Report
Overall, this is a comprehensive and detailed assessment of previous studies, which have sought to develop anthropometric equations to determine height. The methods used were appropriate and the conclusions drawn reflect the findings and I believe this study makes a significant contribution to the field, particularly in how it draws attention to the limitations of previously published studies.
I have provided a few comments below for the authors to consider, which I believe will improve the work and make this manuscript suitable for publication:
- The authors provide good levels of detail about the search strategy used in the systematic review, however, to allow the reader to fully understand the process they should consider including their full search strategy/history as a supplementary file.
- Line 141- the author states that they selected equations that were the most straightforward. What criteria was used to determine if an equation was straightforward, number of predictors?
- Results - you suggest that the values reported in study ID 15 are likely incorrect. What was it that lead you to believe this was the case?
- Results, section 3.2 - I understand that you performed combined analysis of a selected number of studies, however, was it not possible to include equations which used other anthropometric variables but still had the same form (e.g. paper ID 18 (arm span))?
- Your analysis has correctly identified differences in allometric relationships between males and females. Some further discussion of this finding would be useful, specifically around sexual dimorphism and why these allometric relationships differ between men and women.
- The abbreviation of pure error changes from 'EP' in line 118 to 'PE' in line 281 - the author should ensure this is consistent throughout.
- Line 393 - I believe a full stop has been used instead of a comma by accident and should read: accessible anthropometric lengths, especially in adults, aged 60 years and over.
Author Response
Overall, this is a comprehensive and detailed assessment of previous studies, which have sought to develop anthropometric equations to determine height. The methods used were appropriate and the conclusions drawn reflect the findings and I believe this study makes a significant contribution to the field, particularly in how it draws attention to the limitations of previously published studies.
I have provided a few comments below for the authors to consider, which I believe will improve the work and make this manuscript suitable for publication:
Thank you for your comments; we hope to respond appropriately, accepting the present manuscript for publication.
- The authors provide good levels of detail about the search strategy used in the systematic review, however, to allow the reader to fully understand the process they should consider including their full search strategy/history as a supplementary file.
- R: Thanks for your suggestion. Table 1 (Natural history of the project and manuscript) was added to the manuscript.
- Line 141- the author states that they selected equations that were the most straightforward. What criteria was used to determine if an equation was straightforward, number of predictors?
- R Thank you. The corresponding information was added: lines 165-167.
- Results - you suggest that the values reported in study ID 15 are likely incorrect. What was it that lead you to believe this was the case?
- R: Dear reviewer, there is a probability that these data are false, in my humble opinion. They calculate an n of 136, but when misapplying it, they divide it into two groups (136 men and 65 women). Strangely, they present incredibly high correlation statistics (0.99), with almost zero random error in their figures; however, the variance of the estimators is between 6 and 7 cm. They do not present the ANOVA, the changes in R2, or the prediction errors. The slopes of the figures are null or very low; however, in their equation, they are 50% or higher, with a p-value <0.001. I mean, the whole manuscript is very odd.
- Results, section 3.2 - I understand that you performed combined analysis of a selected number of studies, however, was it not possible to include equations which used other anthropometric variables but still had the same form (e.g. paper ID 18 (arm span))?
- R: Dear reviewer. Like you, we understand the need to have valid equations for other anthropometric variables, but according to the strict methodology described in this manuscript, most of the reported equations did not meet the validity criteria or were not validated. On the other hand, as we mentioned in methods, when an author reports several equations, those with the highest adjustment factor (R2, adjusted R2) and the lowest prediction errors are presented.
- Your analysis has correctly identified differences in allometric relationships between males and females. Some further discussion of this finding would be useful, specifically around sexual dimorphism and why these allometric relationships differ between men and women.
- R: Thanks for your suggestion. A line about the meaning of allometric relationships in the regression was added (lines 389-391), also a paragraph in the discussions: (lines 397-404).
- The abbreviation of pure error changes from 'EP' in line 118 to 'PE' in line 281 - the author should ensure this is consistent throughout.
- R: Thank you for your comment. Several writing errors were corrected in this new version
- Line 393 - I believe a full stop has been used instead of a comma by accident and should read: accessible anthropometric lengths, especially in adults, aged 60 years and over.
- We appreciate your comment. The mistake was corrected.
Dear editor and reviewers. In this edition of the manuscript, the corresponding figure 4, already mentioned in the previous version of the manuscript, was added.

Reviewer 2 Report
The authors aimed (i) to make a comprehensive systematic review of manuscripts where equations were designed to predict height by anthropometric methods in adults ≥ 60 years and (ii) to analyze the level of precision with which said proposed equations estimate height.
Overall, this is a very interesting systematic review. However, there are some issues that need to be addressed.
Specific comments
Title
Replace with "...adults: A Systematic Review".
Abstract
The abstract is not in line with the PRISMA 2020 guidelines for abstracts: http://www.prisma-statement.org/documents/PRISMA_2020_abstract_checklist.pdf
Methods
Protocol must be registred earlier, and the PRISMA 2020 guidelines must be strickly followed, namely in the headings.
Elegibility criteria must be followed using PICOS.
Add table for the elegibility criteria.
Scopus and Web of Science why were they not searched?
Author Response
The authors aimed (i) to make a comprehensive systematic review of manuscripts where equations were designed to predict height by anthropometric methods in adults ≥ 60 years and (ii) to analyze the level of precision with which said proposed equations estimate height.
Overall, this is a very interesting systematic review. However, there are some issues that need to be addressed.
Specific comments
Title
Replace with "...adults: A Systematic Review".
R: Dear reviewer, we changed older adults for adults ≥ 60 years in the title.
Abstract
The abstract is not in line with the PRISMA 2020 guidelines for abstracts: http://www.prisma-statement.org/documents/PRISMA_2020_abstract_checklist.pdf
R: Dear reviewer. The summary was modified according to the PRISMA 2020 guidelines
Methods
Protocol must be registred earlier, and the PRISMA 2020 guidelines must be strickly followed, namely in the headings.
R: Dear reviewer. Unfortunately, the protocol was not registered due to time and COVID-19 issues. However, as can be seen, it was methodologically designed by all the six authors together at the beginning of a summer of research. The natural history of the project is summarized in Table 1. Missing headings are also placed in each section of the manuscript
Elegibility criteria must be followed using PICOS.
R: Dear reviewer. The eligibility criteria of the manuscript were modified. We hope that the modification made, together with the added Table 1, will answer that question and the next one.
Add table for the elegibility criteria.
R.; above answered
Scopus and Web of Science why were they not searched?
R: Dear reviewer. The institutional integrators "UACJ and CONRYCYT" already contain the editorial publishers mentioned above.
Dear editor and reviewers. In this edition of the manuscript, the corresponding figure 4, already mentioned in the previous version of the manuscript, was added.

Reviewer 3 Report
The two purposes of this work are: to 80 make a comprehensive systematic review of manuscripts where equations were designed to predict height by anthropometric methods in adults ≥ 60 years and to analyze the level of precision with which said proposed equations estimate height.
A manuscript with a great deal of statistical reflection and good discussion of the existing literature on the subject.
Here are my contributions:
- Include references for the statements in line 37.
- In lines 67, 71, 75, 79… it talks about the height, or the maximum height as above in the text? Clarify the use of the concept of height and maximum height.
- Why was the search not carried out in search engines such as web of science or scopus?
- Specify the reasons for the excluded items in the "included" section of PRISMA. What were the reasons for the exclusion of these 98 items?
- Wouldn't it make more sense to talk about the limitations and strengths of the study after discussing the results?
- Review the format of the references. Some references have different formats, e.g.: 10, 45...
Author Response
The two purposes of this work are: to 80 make a comprehensive systematic review of manuscripts where equations were designed to predict height by anthropometric methods in adults ≥ 60 years and to analyze the level of precision with which said proposed equations estimate height.
A manuscript with a great deal of statistical reflection and good discussion of the existing literature on the subject.
R: Thank you very much for your comment
Here are my contributions:
- Include references for the statements in line 37.
- R: The respective references were included.
- In lines 67, 71, 75, 79… it talks about the height, or the maximum height as above in the text? Clarify the use of the concept of height and maximum height.
- R: Thanks for the clarification. This correction was made where appropriate throughout the manuscript.
- Why was the search not carried out in search engines such as web of science or scopus?
- R: The institutional integrators "UACJ and CONRYCYT" already contain the editorial publishers mentioned above.
- Specify the reasons for the excluded items in the "included" section of PRISMA. What were the reasons for the exclusion of these 98 items?
- R: Thanks for his watching. This explanation was clarified: lines 123-129.
- Wouldn't it make more sense to talk about the limitations and strengths of the study after discussing the results?
- R: Dear reviewer, we ask you to leave us this format in the initial discussions, which allows us, at first hand, to visualize our results, and later discuss their importance, strengths, and weaknesses.
- Review the format of the references. Some references have different formats, e.g.: 10, 45...
- R: Thank you, all of them were reviewed and corrected.
Dear editor and reviewers. In this edition of the manuscript, the corresponding figure 4, already mentioned in the previous version of the manuscript, was added.

Round 2
Reviewer 2 Report
Thank you for your replies!